# Spatial Analysis of the Occurrence and Spread of Wildfires in Southwest Madagascar

**Laura Champin [1,2,\*], Aude Nuscia Taïbi [1] and Aziz Ballouche [1]**

1    Department of Geography, Université d'Angers, ESO Angers UMR CNRS 6590, 5 bis Bd. Lavoisier, CEDEX 01,
     49045 Angers, France; audenousia.taibi@univ-angers.fr (A.N.T.); aziz.ballouche@univ-angers.fr (A.B.)
2    Department of Geography, Université d'Angers, LETG Angers UMR CNRS 6554, 11 Bd. Lavoisier, CEDEX 01,
     49045 Angers, France
\*    Correspondence: laura.champin@univ-angers.fr

**Abstract:** The island of Madagascar, located in the southern hemisphere between the equator and the Tropic of Capricorn in the Indian Ocean, 450 km from the African continent, is particularly affected by wildfires. The vegetation of the phytogeographic Western Domain of the island consists largely of savannas, wooded grassland, and secondary grassland, maintained by the repeated action of fire operating each year on a seasonal cycle. Rural populations employ fire to manage land use. Depending on the burning practice and the environment in which the fires happen, the impacts vary. This paper supplement the studies that have so far located and quantified wildfires by taking into account their different behaviors, particularly their spread. We analyzed the modalities of the relationship between the two fire products, active fire and burned area, derived from Moderate Resolution Imaging Spectroradiometer (MODIS) data to establish a typology based on fire spread patterns. We identified three general patterns of fire behaviors, as well as their locations in the studied area. Spatial analysis of this patterns enabled us to understand spatial logics better. Type 1 fires are the least frequently observed and have many active fires, but little or no burned area. Type 2 fires are the most common and have areas that burn like a mosaic. Type 3 fires are observed slightly less frequently than the previous type and have few active fires and large burned areas. An inter-annual analysis reveals the spatial stability or variability of these fire types.

**Keywords:** spatial analysis; GLM; wildfire; MODIS; Madagascar





## 1. Introduction

The island of Madagascar is almost fully covered by fires [1,2], with nearly 4 million hectares burned each year (mean of 2001–2018), (Figure 1). These fires occur most frequently on the western side of the island, particularly during the dry season, which runs from May through to November [3]. These fires largely correspond to the definition of "bushfires" proposed by Fournier et al. [4], affecting the savannas. However, as is the case in many African nations, the Malagasy authorities have a very strict policy against "bushfires". For the country's environmental management agencies, particularly the forestry department, efforts to suppress such fires are primarily justified by the detrimental impact they have on the environment [5]. Nevertheless, fire is widely employed to manage land use by rural populations, with a diverse array of practices employing various different types of fire. Rajaonson et al. [6] have established a typology of these fires, categorized on the basis of the intentions behind them. Depending on the practices employed and the nature of the fire, the behavior and environmental impact of this burning can vary considerably. In addition to climate factors, topographic variables, such as altitude, slope, and aspect, can also affect the behavior and speed of forest fires' spread [7].

The pastoral burning used by livestock farmers leads to large-scale fires used to increase the quality and quantity of pasture land, stimulating grass growth. They also

serve to reduce bush cover. These fires can range over large areas, clearing space and removing view obstacles, making it easier to keep track of grazing herds. "Dahalo" (zebu thieves) fires are connected with insecurity, opening up passages in the savanna which allow for easy escape. Agricultural fires, meanwhile, are restricted to specific plots of land, as are clearing fires linked to land appropriation and fires set to produce charcoal. "Burning off" fires are slightly more complex. They may be restricted to small piles of dry material in direct proximity to residential areas or farmland, or else they may be used to clear out a whole plot of land, or an even larger area. The areas cleared by such fires thus vary considerably. We might also cite the fires set in the hunting and capture of wild zebus [8], or the mining-related fires set to locate lodes and deposits of precious and semi-precious stones.

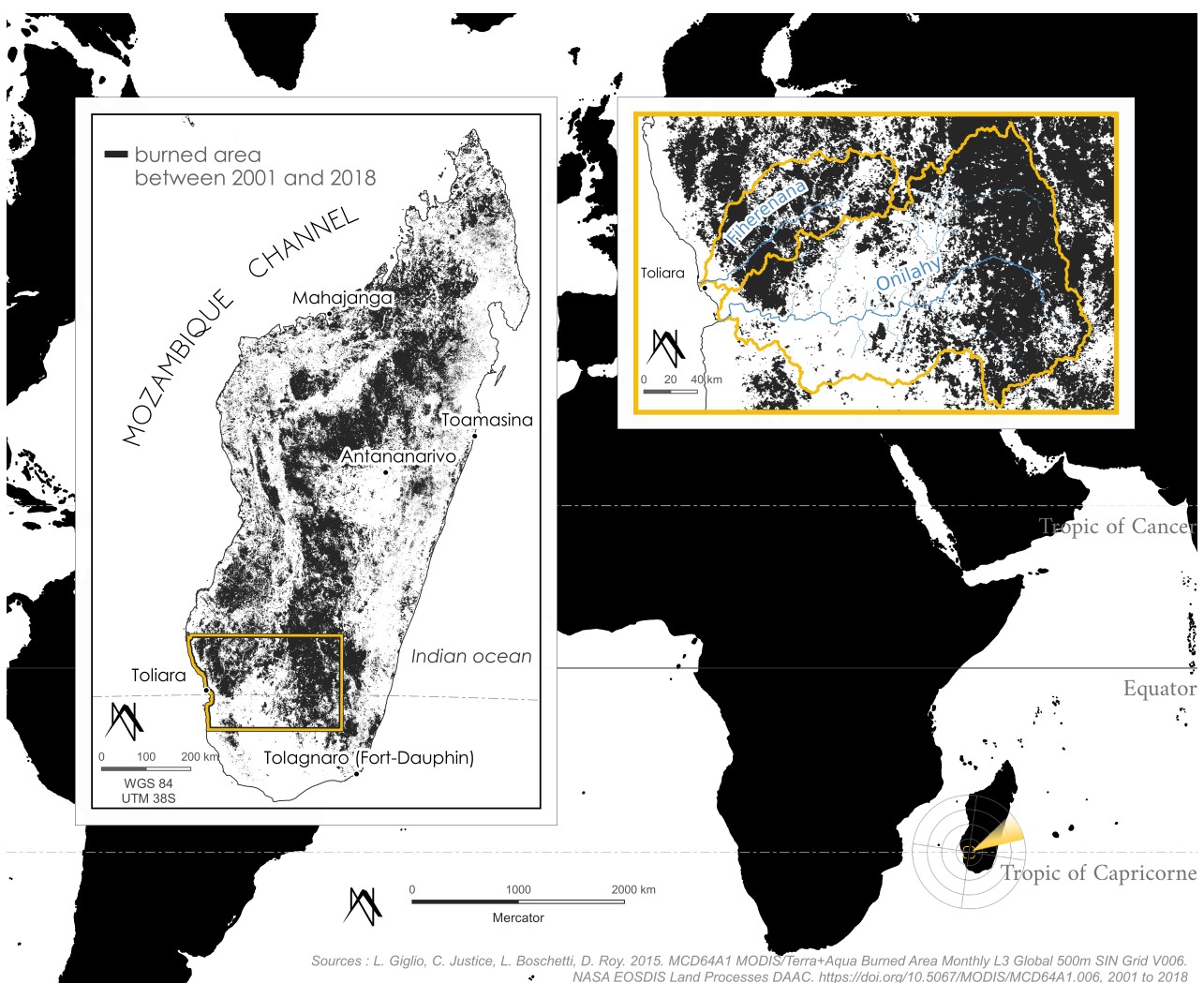

**Figure 1.** Background: Location of the study area in the world, the southwest of Madagascar. Foreground-**left**: Burned areas at least once in the period 2001–2018 in Madagascar. Foreground-**right**: Burned areas in the Fiherenana and Onilahy watersheds, southwestern Madagascar.

In the general literature, fires are often regarded as a major factor in the deterioration of the environment, particularly flora [9]. They are accused of destroying forests, causing a serious decline in biodiversity and leading to erosion and soil loss. In many cases, these processes have been reasonably well-documented in scientific studies, but research has also demonstrated how the use of fire is an integral part of land management, as well as being governed by socio-cultural traditions [5,8]. For example, "doro tanety", a type of agricultural and pastoral fire used in savanna zones, has been described as practical,

protective, and useful [6]. Other studies have illustrated the historic roots of fire and its uses in the Malagasy environment [10–13].

In a bid to clarify the ambiguous status of wildfires in savanna landscapes, much research, particularly in Africa, has sought to identify burned areas, connect that data with plant cover, and analyze the cycles at work, looking in particular at their interannual frequency and whether they occur earlier or later in the year [3,14–22]. Among other sources, this research is based on data derived from remote sensors. Due to the important role that fire plays in the country, Madagascar provides the ideal terrain in which to pursue such studies in greater depth. In Madagascar, the forestry agents of the former Water and Forestry Directorate (DEF), now the Regional Directorate for the Environment, Ecology, and Forests (DREFF), gather information on the location and extent of wildfires, and the nature of the material consumed by these fires. These data are gathered in the field, with all of the constraints associated with the territory, and are not sufficiently exhaustive to give a comprehensive, satisfactory overview of fire systems. This creates real problems with regard to data representativity, and the validity of drawing general conclusions for the whole territory from the cases thus reported is doubtful [23]. Satellite estimates are far superior to those derived from official statistics [24]. In order to overcome these problems, we make use of fire data derived from satellite images, which allow for an overhead view spanning the entirety of the zone in question. Nonetheless, satellite data are not without their own problems and lacunae.

In the 1980s, the first fire detections were made using satellite data from sensors originally designed to observe and forecast meteorological events [25]. Sensors with infrared channels have proved to be very useful for fire detection [26,27]. Since 1999, with the launch of Terra satellite of the Earth Observing System (EOS) program, the moderate resolution imaging spectroradiometer (MODIS) is used to monitor and characterize the fire processes [28,29]. Other authors exploited the 15 min temporal resolution of Spinning Enhanced Visible and Infrared Imager (SEVIRI), aboard Meteosat Second Generation (MSG) geostationary satellites, to monitor fires with an increased frequency of observation. More recent instruments, such as Visible Infrared Imaging Radiometer Suite (VIIRS) and Sea and Land Surface Temperature Radiometer (SLSTR) offering data at higher spatial resolution in the infrared bands (e.g., up to 375 m for VIIRS), have further improved the detection capabilities of active fires [30–32]. Moreover, thresholding methods based on temperature or photosynthetic activity measurements can lead to over- or underestimation errors. For example, in tropical areas subject to alternating wet and dry seasons, photosynthetic activity decreases during the dry season. Thus, an area may be identified as having burned when in fact it has not. The temperature of a fire is not the same depending on the type of vegetation burned. Fires burning a grassy stratum are lower in temperature than fires burning woody vegetation. In addition, the presence of clouds, which are particularly prevalent during the wet season, can compromise the reading of data from optical sensors [33,34].

Errors of commission and omission are a recurring problem for algorithms/products using satellite data to investigate and detect fires [35]. With regard to MODIS fire data in particular, frequently used in small-scale contexts, the 500 m resolution threshold for "burned areas" runs the risk of over- or underestimating the surface area of burned vegetation. Underestimates, caused by errors of omission, may also arise from the algorithm used to identify burned areas, which tends to count only those pixels which it considers to be "burned" with a high degree of certainty, thus failing to count many burned pixels [36]. Problems also arise when the spatial distribution of the fire is too small or too fragmented to be identified as burned vegetation in the satellite images [37]. Clouds and thick smoke can obscure large fires [38]. Overestimation of burned areas, caused by errors of commission, may arise from the fact that individual pixels are counted as entirely burned up, even if the burned surface on the ground is actually smaller than the size of a pixel [39]. To prevent the occurrence of false alarms caused by bright/reflective surfaces (e.g., metal factory rooftops), the MODIS algorithm tries to account for the effects of sun glint [40]. Hot volcanic features and gas flares may also lead to false positives.

Due to the aforementioned issues, these biases that negatively impact data quality make it difficult to precisely determine the number of active fires and hectares burned. The method proposed in this paper does not require exhaustive or even very precise data on fires and burned areas. In this work, we combine information on active fires and burned areas to identify different fire spread patterns in the South-western Madagascar. By analyzing the modality of the relationship between the number of pixels of active fires and the number of pixels of burned areas over a given region, we can identify different fire spread patterns in Sections 2.4.1–2.4.5. Based on these spatial patterns we can establish a typology of different modes of fire spread on portions of regions in Section 2.4.4. This approach allows us to analyze not only the spatial dynamics of wildfires, but also their temporal variation in South-western Madagascar in Section 2.4.6. To this end, our analysis spans an extended period ranging from 2001 to 2018. We begin by identifying the different behaviors of fire observed during this 18-year period in Section 3.1, before focusing specifically on areas in transition, i.e., those areas where the pattern of wildfires has changed most dramatically in Section 3.2. We will then be able to compare these changes with any changes in land use. This step is currently underway and will be the subject of a future publication.

## 2. Materials and Methods

### 2.1. Study Region

The area covered by our study spans 75,947 km² in South-western Madagascar. This region includes the watersheds of the Fiherenana and Onilahy rivers (Figure 1). This relatively large area spans multiple climates. The southern and western zones have an arid climate, the northern zone has a tropical climate and the easternmost zone has a temperate climate (Figure 2) [41].

Average annual rainfall in the north of the western regions is 2000 mm/year, which falls to just 500 mm/year in the south. Average annual temperatures here fluctuate between 25 and 27 °C. The dry season lasts for 7 months, on average. The Southern region is drier, with an average annual rainfall of 300–500 mm/year and a longer dry season of up to 8 months, which has been known to go on for as long as 12 or even 18 months. The east of the Onilahy drainage basin also includes a portion of the eastern region, characterized by higher altitudes and much greater precipitation, with average annual rainfall of 1300 to 1500 mm/year and a shorter dry season of five to six months. Temperatures here are also lower than in the west, with annual averages of between 17 °C and 20 °C [42].

South-western Madagascar is mostly part of the "western" region in terms of flora, with 80% of its surface area covered by various types of savanna, secondary grassland, or wooded grassland, as reported by White [42]. There are also two main types of primary vegetation in South-western Madagascar: dry deciduous forests and deciduous thicket. Dry deciduous forests are very localized, or else form patchwork formations with the savannas. More often than not, they seem to be transitional with thicket land. Deciduous thicket generally exists on the fringes of dry forests. The transition between the two is often very gradual. Thicket results from ecological, climatic, or edaphic conditions, which are not conducive to the development of forests, resulting in a predominantly xerophytic vegetation characterized by high numbers of succulent plants, particularly Didiereaceae and tree species of Euphorbiaceae [43].

As in other tropical regions, fire is the variable which best explains the coexistence of trees and grasses [44]. The frequent burning leads to wide-open savanna landscapes [45]. The wildfire season here stretches from May through to November, broadly corresponding to the dry season [3].

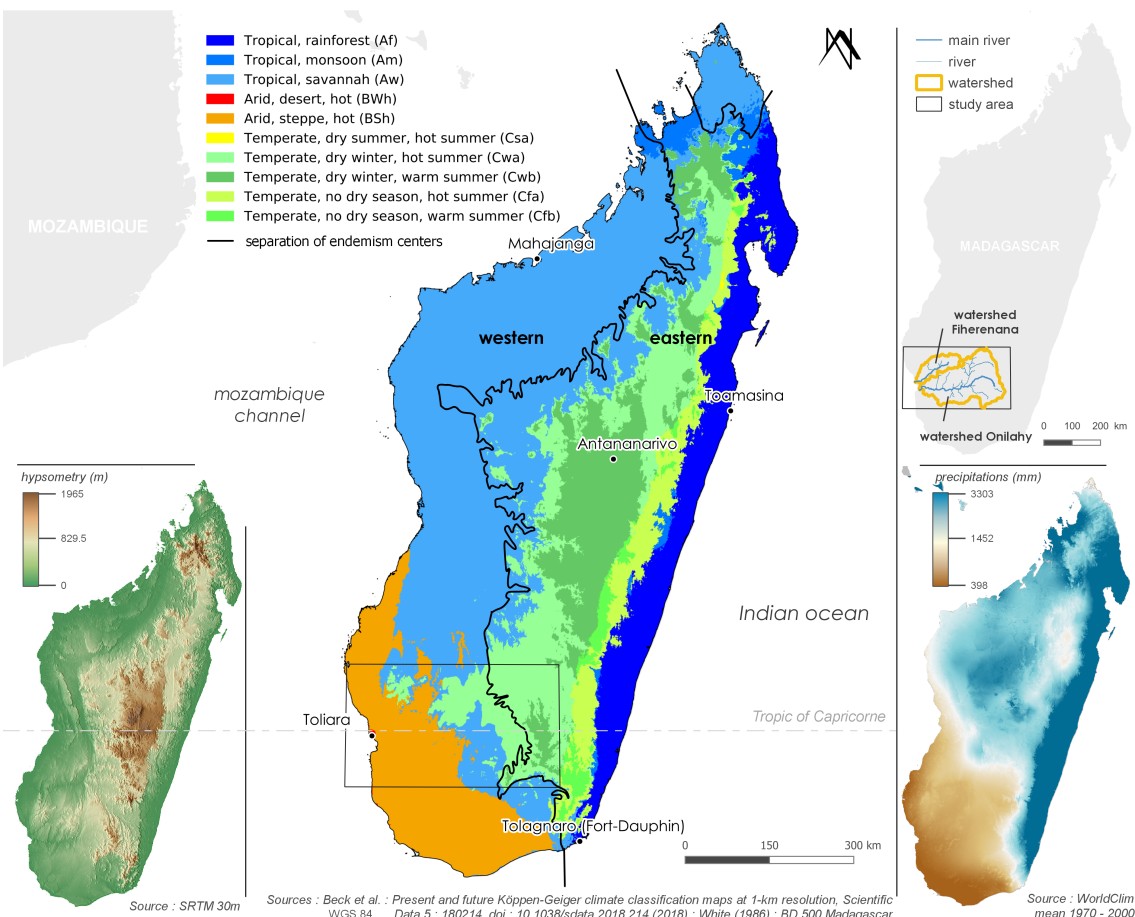

**Figure 2.** Fire predisposition factor, the climate. **Center**: From the Köppen-Geiger climate classification map for Madagascar (1980–2016) and limit centers of endemism from White (1986). **Left**: Hypsometry from SRTM 30 m. Top-**right**: location of watersheds in southwestern Madagascar. Bottom-**right**: Mean rainfall in mm per year between 1970 and 2000.

## 2.2. Data Used

In this study, we used both Aqua and Terra-MODIS data. The Terra satellite crosses the equator at 10:30 a.m. (local time) on a descending node, and Aqua passes the equator at 1:30 p.m. on an ascending node. The MODIS sensors have a viewing swath width of 2330 km, thus providing a complete image of the surface of the Earth every one to two days, in 36 spectral bands ranging from 0.4 µm to 14.4 µm. The spatial resolution varies depending on the spectral bands: 250 m for bands 1 and 2, 500 m for bands 3 to 7, and 1000 m for bands 8 to 36.

The MODIS data are used to generate "fire products"—active fires and burned areas. User manuals are available for both the "active fire" and "burned area" products [46,47].

These data present a number of advantages for our study. The temporal resolution of the data—grouped into 8-day periods and available from the 2000 s onwards—devoted to interannual analysis in order to track developments over a relatively long period of time. The spatial resolution is also the most suitable option for the scale of our analysis, which spans some 75,947 km². Our aim is to identify fire systems on a small scale, and study their specific patterns. Both products "active fire" and "burned area" products can be complementary [17]. By analyzing the two fire products, we can ascertain patterns of fire behavior.

The "Fire products" used in our analysis are available online: https://ladsweb.modaps.eosdis.nasa.gov/ (accessed on 22 May 2022). For "active fire" products, we downloaded the products MOD14A2 (Terra) and MYD14A2 (Aqua) in raster format GeoTIFF. The same format was selected for "burned areas" product MCD64A1.

The processing algorithms and daily data output for the planet as a whole are different for the "active fire" and "burned area" products [48].

### 2.2.1. Active Fire Products

Fire Products: MOD14A2 (Terra) and MYD14A2 (Aqua) Level 3 include 46 raster images per year (8-Day Summary), in which each pixel is assigned to a class (Table 1), 3 of which (classes 7, 8 and 9) indicate the presence of fire, with varying degrees of confidence. The finished product detects fires burning at the time of overpass under relatively cloud-free conditions, the energy emitted and the flaming-to-smoldering ratio, for pixels with a resolution of 1 km [49].

**Table 1.** MOD14/MYD14 fire mask pixel classes, from Giglio (2020).

| Class | Signification |
|---|---|
| 0 | not processed (missing input data) |
| 1 | not processed (obsolete; not used since Collection 1) |
| 2 | not processed (other reason) |
| 3 | non-fire water pixel |
| 4 | cloud (land or water) |
| 5 | non-fire land pixel |
| 6 | unknown (land or water) |
| 7 | fire (low confidence, land or water) |
| 8 | fire (nominal confidence, land or water) |
| 9 | fire (high confidence, land or water) |

The algorithm is entirely automated, and builds on the heritage approach used for the Geostationary Operational Environmental Satellites (GOES) and Advanced Very High Resolution Radiometer (AVHRR) sensors [29,50,51]. It is used to process MODIS images from both Terra and Aqua, where the apparent temperature is observed from band 22 upwards (or band 21 if 22 is saturated or data is missing), from 3.929 to 3.989 μm and band 31 from 10.78 to 11.28 μm. Band 2, in the near-infrared band of 841 to 876 nm (Table 2), is also used to identify highly reflective surfaces, liable to be mistaken for fires and thus provoke false reports [48].

The first processing step eliminates all pixels not considered to represent potential fires, with a temperature threshold judged to be too low, and adapted for day and night. The detection of fire is then determined by absolute fire detection, which is to say areas where the temperature is above the threshold values set for day and night in band 22, or else above a specific threshold for the value of band 22 minus band 31. This absolute detection only concerns fires of sufficient intensity. For low intense fires, detection depends on the thermal emissions from surrounding pixels. Pixels representing high temperatures are no longer identified using an absolute threshold, but instead in relation to the background thermal emissions detected from the surrounding pixels [49].

**Table 2.** MODIS bands (in red, used by the active fire detection algorithm).

| Bands | Wavelengths | Spatial Resolution |
|:---:|:---:|:---:|
| 1 | 620–670 nm | 250 m |
| 2 | 841–876 nm | 250 m |
| 3 | 459–479 nm | 500 m |
| 4 | 545–565 nm | 500 m |
| 5 | 1230–1250 nm | 500 m |
| 6 | 1628–1652 nm | 500 m |
| 7 | 2105–2155 nm | 500 m |
| 8 | 405–420 nm | 1000 m |
| 9 | 438–448 nm | 1000 m |
| 10 | 483–493 nm | 1000 m |
| 11 | 526–536 nm | 1000 m |
| 12 | 546–556 nm | 1000 m |
| 13 | 662–672 nm | 1000 m |
| 14 | 673–683 nm | 1000 m |
| 15 | 743–753 nm | 1000 m |
| 16 | 862–877 nm | 1000 m |
| 17 | 890–920 nm | 1000 m |
| 18 | 931–941 nm | 1000 m |
| 19 | 915–965 nm | 1000 m |
| 20 | 3.660–3.840 µm | 1000 m |
| 21 | 3.929–3.989 µm | 1000 m |
| 22 | 3.929–3.989 µm | 1000 m |
| 23 | 4.020–4.080 µm | 1000 m |
| 24 | 4.433–4.498 µm | 1000 m |
| 25 | 4.482–4.549 µm | 1000 m |
| 26 | 1.360–1.390 µm | 1000 m |
| 27 | 6.535–6.895 µm | 1000 m |
| 28 | 7.175–7.475 µm | 1000 m |
| 29 | 8.400–8.700 µm | 1000 m |
| 30 | 9.580–9.880 µm | 1000 m |
| 31 | 10.780–11.280 µm | 1000 m |
| 32 | 11.770–12.270 µm | 1000 m |
| 33 | 13.185–13.485 µm | 1000 m |
| 34 | 13.485–13.785 µm | 1000 m |
| 35 | 13.785–14.085 µm | 1000 m |
| 36 | 14.085–14.385 µm | 1000 m |

2.2.2. Burned Area Products

The burned area mapping algorithm handles the MODIS imagery with a spatial resolution of 500 m, seeking to identify persistent changes caused by the combustion of biomass in pixels. The daily surface reflectance values in Bands 5 (1230 to 1250 nm) and 7 (2105 to 2155 nm) atmospherically corrected are employed to derive a burn-sensitive vegetation index (VI). Composite imagery summarizes the persistent changes in the time series of the VI. Indeed, the index shows a significant decrease rapidly after burning. In this step, the absence of clouds on the reflectance values is verified with the internal cloud mask (MO/YD09) and the absence of active fires with the MODIS active fire observations (MO/YD14). A cumulative mask is processed with active fire data to guide the selection of burned and unburned training samples and specify the fire date in each grid cell. The algorithm then performs a classification based on training samples of burned and unburned areas. Burned training grid cells are created from the cumulative mask of active fires and to reduce the impact of under-sampling of spatial fire extent by active fire observations. It is expanded through a process of region growing, constrained by a maximum growth distance of 10 km. Unburned training grid cells must meet the following selection criteria: they must be "a-priori" unburned, i.e., identified as unburned based on the VI time series and they must contain valid data, i.e., not be in the burned training mask. This MODIS burned area collection 6 is described in Ref. [47] and the production algorithm is fully detailed in Ref. [52].

2.2.3. Differences between the Data for These Two Products

The main difference between the "active fire" and "burned area" products is the ephemeral nature of active fires and the lasting changes to burned areas. Fires of short-term duration may be undetected by MODIS owing to the daily temporal coverage. Burned areas, on the other hand, can be observed over longer periods of time, since it takes more than two days for vegetation to grow back. Repeated flyovers by satellite thus allow the algorithm to identify burned areas which may be concealed by clouds or thick smoke, or which may not have been burning during the first flyover. "This implies that active fire and fire-affected area products may be used in conjunction to improve fire information" [37].

*2.3. Pre-Processing the Data*

All of the active fire and burned area data retrieved for the period 1 January 2001 to 31 December 2018 were processed using the R Core Team software program [53].

We began with a few pre-processing operations in order to prepare the data for our calculations (Appendix A, Figure A1).

For the MODIS "active fire" data, we conducted the same processing operations for the Terra and Aqua data.

To begin with, the active fire pixels were reclassified: a value of 1 was assigned to all pixels declared to contain an active fire. We selected all fires, regardless of confidence level [54]. NA was assigned to fireless pixels (Class 5 in the MOD14A2 and MYD14A2 data), and a value of 0 for pixels where fire is not possible (Classes 3, 4, and 6 in the MOD14A2 and MYD14A2 data) (Table 1). A distinction is thus made for pixels in Class 5, because even if the algorithm does not detect a fire, these pixels are at least capable of containing fire, compared with Classes 3, 4, and 6, which cannot, because they either represent areas covered by water or else a degree of uncertainty that is too high, for example as a result of cloud cover.

The 46 raster images from a given year were then combined with each pixel given a value (0–46) equal to the number of images in which they were classified as active fire. The active fire pixels, which have an initial resolution of 1 km, were then disaggregated to 500 m, corresponding to the spatial resolution of the pixels used to detect burned areas. Finally, data taken from Terra were combined with those from Aqua to obtain a single raster image of active fires over the course of the year.

The procedure applied to the burned area data is essentially the same, with the exception of the disaggregation operation and the merging of the Terra and Aqua data, since the data are directly provided in a single raster file. We used the data layer "burn data" to which we assigned the value of 1 for each burned pixel, i.e., pixels with values greater than 0 (from 1 to 366). We assigned the value of 0 to the unburned pixels, i.e., those less than or equal to 0, 0 being the unburned land, −1 the unmapped area due to insufficient data and −2 the water surfaces.

### 2.4. Methods

#### 2.4.1. Heatmap of Active Fires and Burned Areas

The first step of data processing involves the spatialization of active fires and burned areas. In order to represent the concentration of active fires and burned areas, we used Quantum Geographic Information System (QGIS) to create a density raster (heatmap) using point vector layers for active fires and burned areas for all years between 1 January 2001 and 31 December 2018. To do this, all raster files for active fires and burned areas derived from the pre-processing phase were combined in advance. Through vectoring, we thus obtained a point file whose value corresponds to the sum total of detected active fires and burned areas. We then calculated a heatmap using the Kernel Density estimation, based on the value of these points for specific locations, and using a radius of 10,000 m and a cell size of 1000 m for the output raster. Since both burned areas and active fires vary from one year to the next, performing this density calculation over the whole period enabled us to broadly determine their spatial distribution.

Our results (Figure 3) reveal a clear difference between the spatial organization of active fires and burned areas. We thus set about questioning and attempting to better comprehend the relationship between these two variables.

#### 2.4.2. Relationship between Active Fires and Burned Areas

We thus set out to document the relationship between active fires and burned areas for each year between 2001 and 2018. Our objective was to determine whether or not there is a linear relationship between fires and burned areas, which could be defined in the form of a pattern.

First, it is important to take into account certain characteristics of the data used, particularly the ephemeral nature of active fires and the persistence of burned areas. The Terra and Aqua sensors are thus capable of detecting a certain number of active fires, perhaps even over several days, which will correspond to a resulting burned area. There will therefore logically be more burned areas than active fires detected, or at the very least an equal number, unless some burned areas prove to be too small for detection [55]. All previous studies utilizing the MODIS "burned area" data have observed a high rate of omission of pixels corresponding to burned areas. However, observing the ratio between the number of active fires observed and the corresponding quantity of burned areas could reveal interesting information, since the passage of the satellite at regular intervals should make this ratio relatively stable. For this reason, we have chosen to observe whole-year data.

By accumulating the data for a whole year, we are left with a certain number of active fires and a certain surface area of burned land. We can identify very extensive burned areas for which few active fires have been detected, which in theory would suggest that the fires spread rapidly and over a great distance. At the other end, when the sensor has not identified any burned areas (as they are too small), but numerous active fires have been reported, we can infer the presence of frequently recurring fires. Analyzing the interrelationship between active fires and burned areas thus allows us to identify different fire systems.

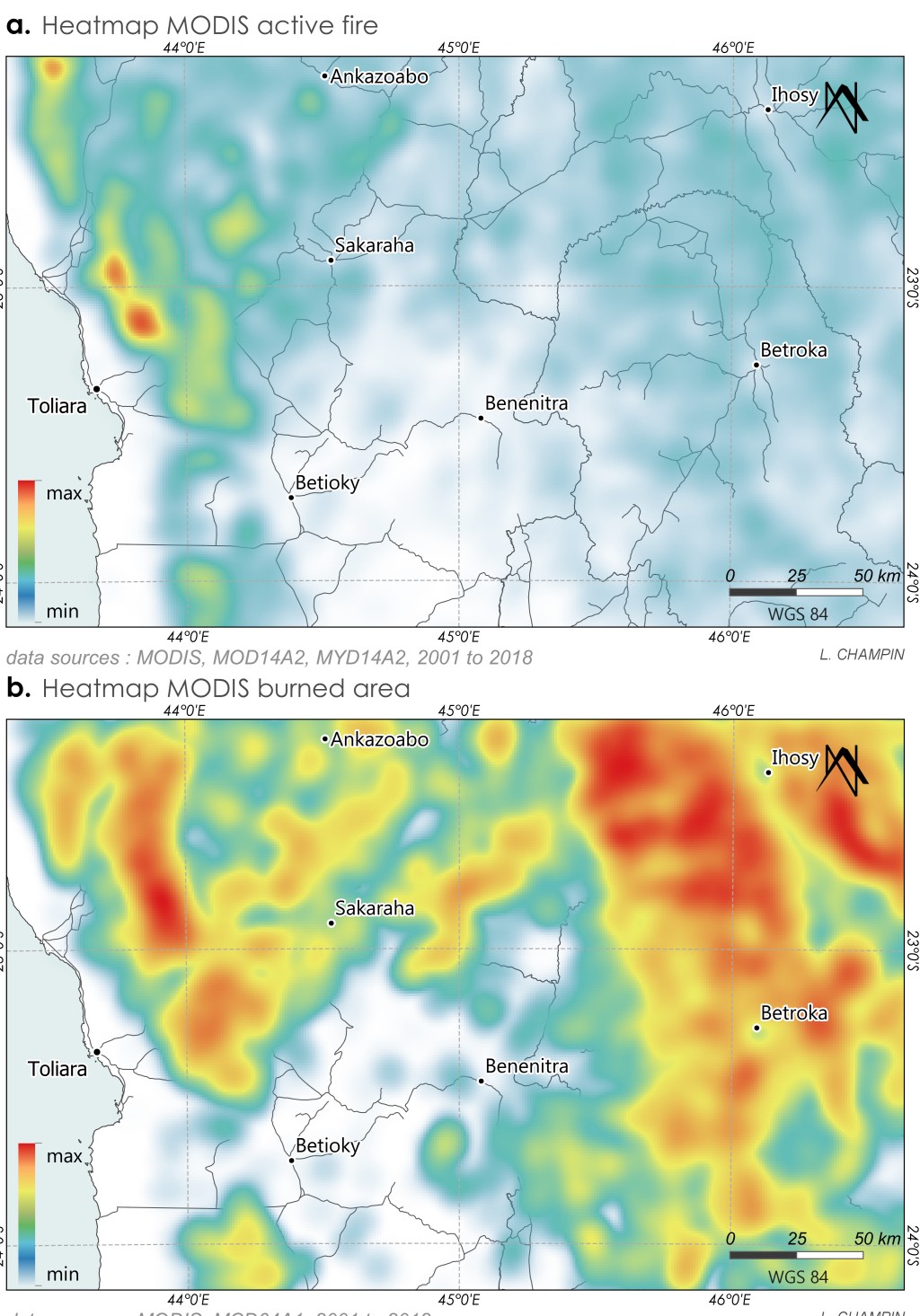

data sources : MODIS, MOD14A2, MYD14A2, 2001 to 2018

data sources : MODIS, MCD64A1, 2001 to 2018

**Figure 3.** Fire density 2001 to 2018. (**a**) Heatmap from the MOD14A2 and MYD14A2 active fire data. (**b**) Heatmap from the MCD64A1 burned area data.

To this end, we use a hexagonal grid with cells of 15 km wide and 17.32 km high (giving a surface area of just under 200 km²). There are 389 cells. The size of these cells—not too small, not too large—is compatible with the area covered by this study, and allows us to identify all different types of wildfire occurrences. The choice of a hexagonal grid allows us to minimize boundary effects. The grid is based on information taken from the raster of active fires and burned areas used in the previous stages, covering the years 2001 through

2018. A function then counts the number of active fire pixels within each cell, then the number of burned area pixels (Appendix A, Figure A2).

The graphs generated by this process (Figure 4) show the relationship between the two variables "active fires" (*x*-axis) and "burned areas" (*y*-axis) for each year. Each point represents a cell within the grid.

For all of the years studied, the scatter plot appears to be split between various different linear relationships. It is quite a challenge to identify and interpret any clear relationship, but it is equally difficult to declare that the two variables are totally independent.

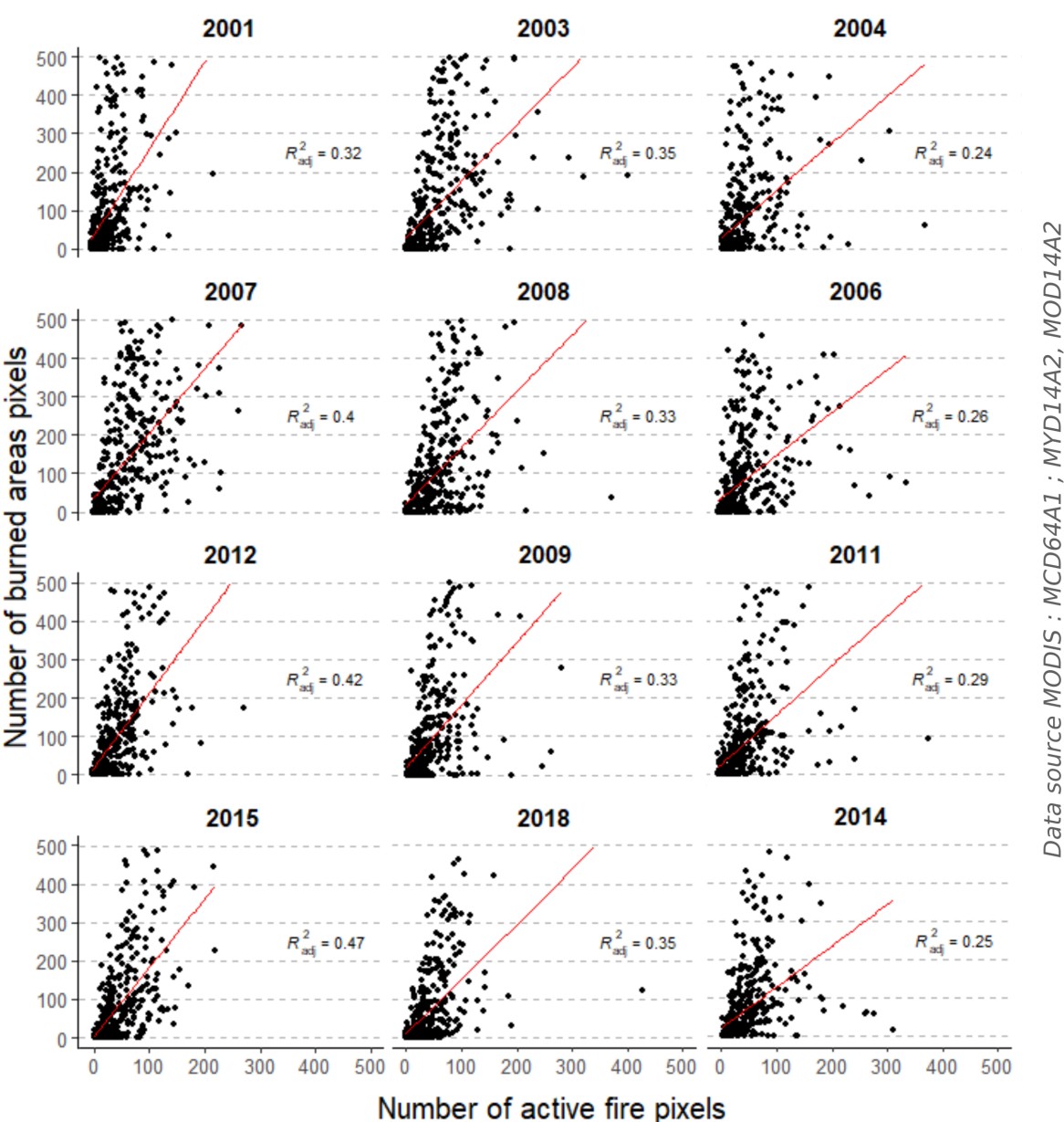

**Figure 4.** Correlation between burned areas and active fires. Example of some years, classified by trend.

### 2.4.3. Measure of Relationship

To assess the independence or intensity of the relationship between the two variables, we thus use a test of independence which allows us to compare the observed frequencies with a theoretical distribution, by measuring the disparity between the two. In this case, the theoretical distribution represents the hypothesis that active fires and burned areas are

entirely independent of one another. To simulate this, we calculate a contingency table for the frequency distribution of the active fires and burned areas, and another, theoretical, contingency table representing active fires and burned areas that are mathematically independent of one another. We then calculate the sum of the deviation between the observed situation and the theoretical situation, in each cell of the contingency table, using the Chi2 or $\chi^2$ method [56]. The result is then compared with a $\chi^2$ table for a risk of error $(\alpha)$ set at 0.05. Since the degree of freedom $(df)$ corresponding to our observations is greater than 30, the corresponding $\chi^2$ is calculated as follows.

$$\chi^2 = \frac{(u + \sqrt{2df + 1})^2}{2} \tag{1}$$

where $u$ = 1.6449 as per the normal distribution function, centered for $(\alpha)$ = 0.05 ; $df$ = degree of freedom.

Since these results rule out independence as a viable hypothesis, it seems pertinent to calculate a coefficient that will allow us to measure the intensity of the relationship between the number of active fire pixels and the number of burned area pixels. Tschuprow's coefficient (T) allows us to measure the strength of the association, where the value indicated is a non-dimensional number between 0 (independence) and 1 (perfect dependence) [56].

$$T = \sqrt{\frac{V}{n * \sqrt{(k-1)(p-1)}}} \tag{2}$$

where $V = \chi^2$ calculated; $n$ = total number; $(k-1)(p-1)$ = number of degrees of freedom.

### 2.4.4. Linear Model and Residuals

Based on these initial results, it appears that there are different types of fire that can be identified with reference to the correlation between active fires and burned areas. To accurately identify and characterize these fires, and establish a typological classification, we will look more closely at the correlation between the number of burned area pixels and the number of active fire pixels detected. The idea is to identify, using a regression line, those cells in the grid which correspond to the linear model derived from the correlation between active fires and burned areas observed in the previous stages, and those which deviate from this model.

To do this, we begin by replacing the active fire and burned area data with their logarithms, which allows us to transform (anamorphosis) the curve into a line. We then calculate the regression of the number of burned area pixels (dependent variable) against the number of active fire pixels (explanatory variable). We use the generalized linear model (GLM) in order to discard the large number of 0 values contained in the statistical series.

Regression is a means of breaking down the information contained in the statistical series. On one side, the regression line represents the estimated variable, where the dependent variable (number of burned area pixels) is linearly dependent in linear fashion on the explanatory variable (number of active fire pixels). On the other side, we have the residuals, which are independent of the explanatory variable. This residual information represents the disparity between the observed situation and the constructed model, which is of interest to us because these residuals inform us of the gap between each cell in the grid and the projections of the model, giving us a measure of specificity (Figure 5).

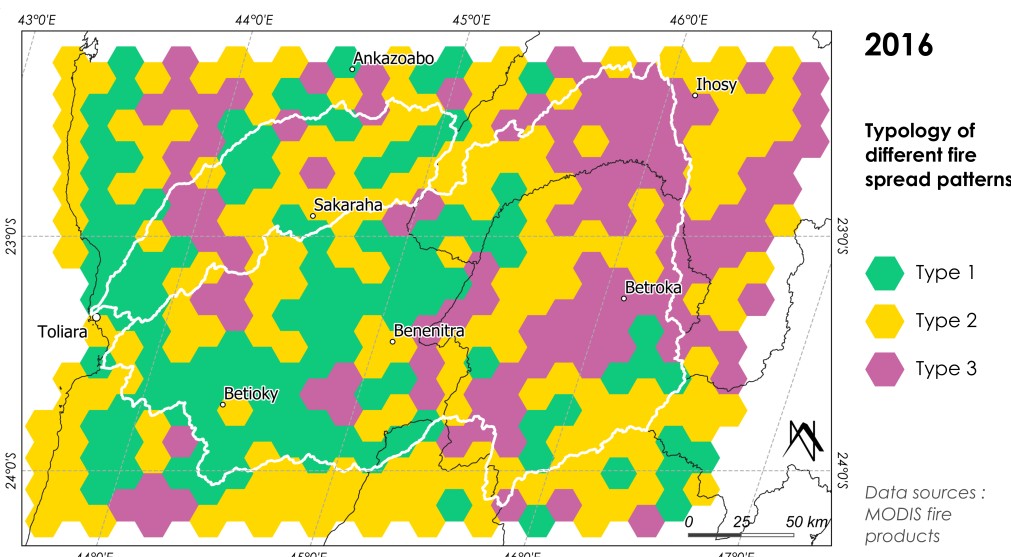

**Figure 5.** Residuals mapping, focus on the year 2016. Each year between 2001 and 2018 are available in (Appendix D).

2.4.5. Coefficient of Determination

Calculating the coefficient of determination allows us to measure the proportion (in %) of information taken into account by the regression model. Since the total sum of distribution information is equal to the information taken into consideration in the regression model plus the residual information, this coefficient of determination allows us to determine how much information corresponds to the model and how much is regarded as residual. The coefficient of determination is equal to the square of the coefficient of correlation.

Taking this analysis further by graphically representing the values in the form of a box plot (Figure 6), showing the number of cells corresponding to each type of fire in a given year, allows us to observe year-on-year variation.

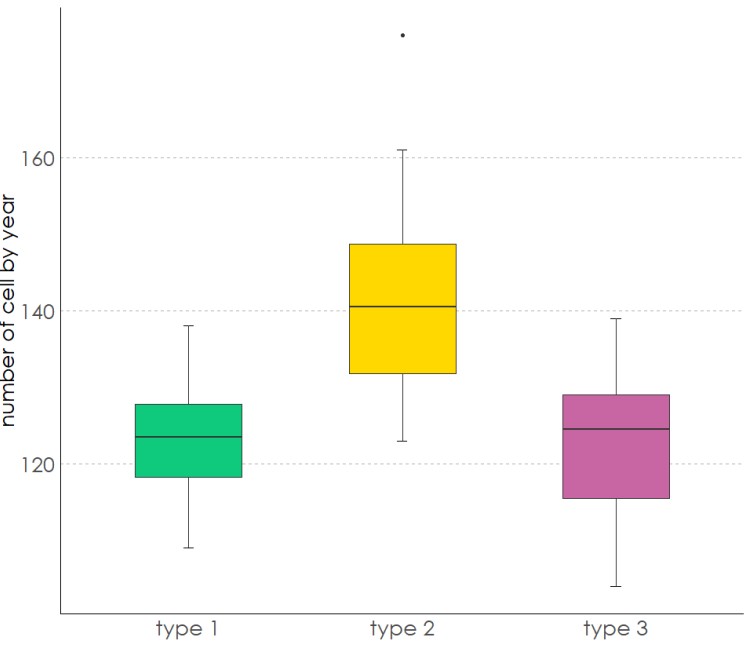

**Figure 6.** Number of cells for each type of fire.

### 2.4.6. Range of Residual Values between 2001 and 2018

To distinguish between areas where fire patterns did not change over the years we studied, and those where change did occur, we extracted the maximum and minimum residual values for each cell in the grid and for each year between 2001 and 2018, then calculated the range between these maximum and minimum residual values. This allowed us to divide the statistical series into four classes, based on mean values and standard deviation, and subsequently map these categories (Figure 7).

Taking this analysis further, for each year in the period 2001–2018 we counted the number of cells corresponding to each type of fire and also falling within the category "fairly stable." We then did the same for those in the "significant change" category. This gave us a table showing, for each type of fire and each year, the number of stable cells and the number of cells experiencing significant change. We then calculated these values as a percentage of the total for each type of fire. This allowed us to compare different types of fires, without worrying about the differences in the number of cells corresponding to each type of fire.

Graphically representing these values in the form of a box plot (Figure 8) allows us to fine-tune the results and the conclusions to be drawn from them.

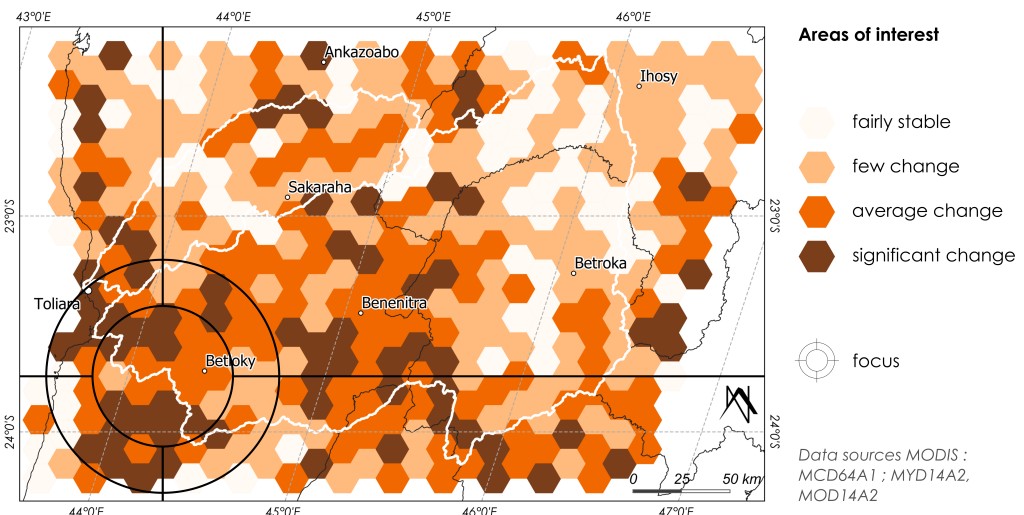

**Figure 7.** Cell-by-cell transformations in fire patterns 2001–2018.

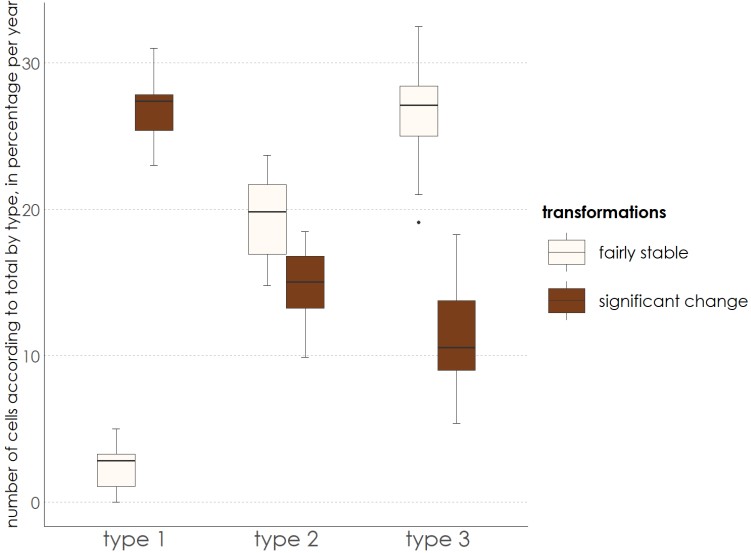

**Figure 8.** Number of cells corresponding to each type, as a percentage of the annual total.

## 3. Results

### 3.1. Spatialisation and Types of Fire

3.1.1. Analysis of Heatmap of Active Fires and Burned Areas

The first observation to emerge from our heatmaps (Figure 3) is that fires occur across nearly the whole study area, the South-western Madagascar, with the exception of its southern tip. This is most likely due to the local climate, which is too arid to provide sufficient combustible material.

The active fire heatmap (Figure 3a) shows two zones at the western and eastern fringes of our area of study that are particularly prone to active fires, with far greater intensity in the west than in the east. These two zones can also be clearly distinguished on the burned area heatmap (Figure 3b), but here the intensity appears to be equal in east and west. Moreover, the high-intensity burned areas in the west cover display spatial disparities with the active fire heatmap. These results show a difference between areas marked by the active fire products and those affected by burned area products, similar to the findings of Devineau et al. [54] in their study in Burkina Faso: "The ability of active fire data and of burned area data to account for areas affected by fire is somewhat different".

3.1.2. Relationship between Active Fires and Burned Areas

As regards the relationship between burned areas and active fires (Appendix C), for each of the years studied, the scatter plots reveal different linear relationships. Two groups show weak correlation. In one, the correlation line is very steep, as visible in the first column of graphs (Figure 4), with a fairly small number of active fire pixels and a number of burned area pixels that varies considerably. This indicates a situation of statistical independence, whereby the number of active fire pixels has little influence on the number of burned area pixels. In the second case, on the contrary, the number of burned area pixels is fairly small while the number of active fire pixels varies considerably, as visible in the third column of the graph (Figure 4). In this case, the number of active fire pixels has little influence on the number of burned area pixels, which remains low regardless of the number of active fire pixels. Between these two extremes, a third trend emerges for the correlation between active fires and burned areas, characterized by a more linear statistical relationship, but one where the number of burned area pixels is always greater than the number of active fire pixels. This case is visible in the second column of the graph (Figure 4).

In all years the calculated $\chi^2$ is greater than the $\chi^2$ values in the table (Appendix B, Table A1), so the hypothesis that the number of active fire pixels does not depend at all on the number of burned area pixels can be rejected. There is indeed a relationship between these two variables.

The calculated Tschuprow coefficient varies between 0.63 and 0.74 for the period 2001–2018 (Appendix B, Table A2), which indicates the existence of a connection, but not a very strong one. This suggests that the number of active fire pixels is just one of several factors which determine the number of burned area pixels.

Drawing upon studies of fire patterns conducted in other parts of the world, we can put forward the hypothesis that the land cover of burned areas is one of those contributing factors that determine the extent of burned areas [54]. In this respect, the work of Monnier [57] is particularly instructive, especially the three conditions required for fire to spread: a sparking event, the presence of combustible material, and the continuity of the latter. This reflects the decisive impact that the nature of vegetation has on the way fires behave.

These results remain fairly ambivalent, with some cases which resemble statistical independence. However, the statistical calculations reveal the existence of a certain relationship nonetheless, albeit a weak one. This suggests that there are different mechanisms at work.

### 3.1.3. Linear Model and Residuals

Residuals are results that are independent of the explanatory variable, i.e., the number of active fire pixels. The further the residual value is from 0, the less likely it will be that the dependent variable (in this case the number of burned area pixels) is determined by the explanatory variable. In short, the further away we move from the linear regression, the less directly the extent of burned areas will depend on the number of active fires detected. A positive residual indicates that the extent of burned areas in the cell in question is higher than the models would lead us to expect, meaning that the number of burned area pixels is greater than expected, while a negative residual indicates a number of active fire pixels greater than that predicted by the model. Residual values close to 0 indicate that the figures correspond closely to the model.

This allows us to split the residual results into three categories, using the first and third quartiles of the statistical series as threshold values (Table 3). These three categories correspond to three distinct patterns of fire spread, which we can map for all of the years in our period (Appendix D).

**Table 3.** Typology of different fire spread patterns.

| Type 1 | (Min–Q1] | Lots of active fires, variable number of burned area pixels |
|--------|----------|-------------------------------------------------------------|
| Type 2 | (Q1–Q3]  | Number of active fire pixels and burned area pixels correspond to regression model |
| Type 3 | (Q3–Max) | Lots of burned area pixels, variable number of active fire pixels |

Mapping these residuals for the whole of South-western Madagascar (Appendix D), we can spatially define the three patterns of fire spread identified in individual cells in the years between 2001 and 2018. The results for the year 2016 are provided as an example (Figure 5).

Although some subtle disparities can be observed over the years analyzed, there is a certain invariance of behavior in space, which allows us to identify two particular zones. Broadly speaking, this mapping exercise allows us to divide the area spanned by our study into two distinct zones. In the east, and in all years, we find many more Type 3 fires, i.e., a small number of detected active fires that leave a large amount of burned land. The west, on the other hand, sees far more Type 1 fires, i.e., a large number of active fires are detected, but they leave behind them little or no burned land. Zones characterized by Type 2 fires, i.e., with a linear distribution between active fires and burned areas, consistent with the regression model, are more or less located between these two main zones.

### 3.1.4. Results of the Coefficient of Determination

According to the coefficient of determination (Appendix B, Table A3), our regression accounts for between 25 and 40% of the total information. This means that 60 to 75% of the information falls into our residual categories. This high proportion of residual information indicates that, in the area covered by our study, many zones function in a manner which does not correspond to the predictions of the model. The large number of residual values, both positive and negative, are poorly represented by the model, and are thus determined by one or more explanatory variables other than the number of active fire pixels, as discussed above.

This distribution can be observed in the box plot showing the number of cells corresponding to each type of fire (Figure 6). It is worth noting that Type 1 fires are the category least frequently observed in the area covered by our study, and that the year-on-year variation in their number is less substantial than that seen in the other categories. Type 2 fires are the most common, and the year-on-year variation in their number is much greater. Type 3 fires are observed slightly less frequently, and their year-on-year variation is also slightly below that recorded for Type 2.

*3.2. Trends in the Development of Different Types of Fire*

Analysis of Residual Values

We built upon this initial observation by calculating and spatializing the range of residual values for each cell in the grid, for all of the 18 years in our period (Figure 7). This allows us to observe the evolution of our different types of fires, and to identify areas where fire patterns have remained stable, i.e., the residual values are stable across the 18 years, and others which have witnessed frequent changes, i.e., where the residual values vary considerably.

Cells identified as stable may correspond and stick to one of the three types of fire defined above , whereas those which vary significantly will have witnessed a change among this fire type.

Two large areas appear to have remained relatively stable, one to the east and the other to the north-west, as well as the whole of the coastline. The southern regions, particularly the south-west, are those where the fire patterns have undergone the greatest change.

## 4. Discussion

After having discriminated three different patterns of fire spread, we will first discuss this typology and its spatial distribution, and then address the dynamics of evolution and potential transformations of the different patterns identified. First, it is necessary to show the relevance of the approach that consists in developing a typology based on the relationship between the number of fire pixels in a given area for each of the fire products.

There is a clear difference between the maps of active fire and burned area products. Although active fire products are present in the burned area processing algorithm, as described by Giglio et al. [52], they serve as a guide for the training samples that support the final burned area classification but do not determine the final burned area detection results. The burned area data retrieved from MODIS imagery are not the result of the accumulation of data from the active fire products, so they can be used to identify unique spatial dynamics and to distinguish areas that behave differently.

### 4.1. Analysis of the Spatial Distribution of Fire Patterns

Previous work [58–61] has already shown that fire use practices and the composition of the landscape can explain the spatial distribution. Based on these results, we can advance the hypothesis that Type 3 fires, with extensive burned areas but only a small number of active fires, appear to correspond to the use of fire by pastoralists, miners, or "dahalo". These fires affect savanna landscapes, with very wide-ranging burning used to clear land and remove obstacles [6,8,23].

The fires identified here as Type 2 leave behind a patchwork of burned areas. The very high number of active fires recorded indicates the continued presence of combustible material in the periods between satellite flyovers. However, fires burning over large areas would not leave any combustible material in their wake. These fires correspond to controlled agricultural fires limited to specific plots of land, or land-clearing and charcoal-making fires, which are also highly localized and limited in scale [62]. They reflect controlled burning practices deployed in strategically selected locations. The composition of some landscapes may also prevent fires from spreading to engulf larger areas [63].

For fires identified as Type 1, where active fires are detected but few or no burned areas are observed, we can surmise that a fire did indeed occur, but that the resulting burned area was too small to be picked up in the 500 m pixels used for MODIS images. For example, the fires used to cook yams directly after harvesting, a fairly widespread practice in forest areas, may generate such results [62]. So too may certain forms of "burning off", especially when the combustible materials are gathered into a single pile before being set alight.

Among the factors which may explain these initial results, the nature of the vegetation present is very important, as discussed above. A lack of dry, combustible material will stop forest fires from spreading, restricting them to the dry vegetation at floor level [8,59,63].

*4.2. Analysis of Dynamics Transformation of Fire Systems*

As numerous studies have already demonstrated, the evolution over time of the distribution of fires and burned area is very instructive [1,20,37,64,65]. Here we have chosen to focus rather on the evolution, not only quantitative but qualitative, of our fire types. We could propose a number of hypotheses to explain the spatial changes in the typology. They might be the result of changing practices, of transformations in the landscape, or both, since the two are closely linked. For example, the transformation of pastures into agricultural land is highly likely to be accompanied by a change in fire use practices, as well as changes to the landscape.

Areas which have seen "significant change" may reflect changes in fire types resulting from the "rotation" of burned areas. We will explore these explanatory factors in our study.

An analysis of these changing fire patterns, focusing exclusively on those cells which have changed significantly and those which have remained stable (Figure 8), demonstrates that those corresponding to Type 1 fires tend to change frequently over the years. Cells characterized by this type of fire show a clear tendency to vary from year to year. Type 2 cells, while more likely to be stable, may also display frequent changes. Cells corresponding to Type 3, however, have a tendency to remain stable from one year to the next.

On this basis, we can further develop our hypotheses regarding the different patterns of fire and their spatial and temporal evolution. Type 1 fires are localized burnings which do not spread. They generally correspond to fires used for cooking or burning off waste. These cooking fires are not domestic in nature, corresponding instead to the fires set in forest areas to cook and prepare produce such as yams. These fires are rarely set in the same place twice, and their location tends to vary from year to year. The fires identified as Type 2 correspond to a patchwork of small burned areas. These patterns are associated with controlled burning, generally limited to specific plots, most often for agricultural purposes. The proportion of such spaces that remain unchanged from year to year, and thus see regular fires, is broadly the same as the proportion of spaces affected by irregular fires, which vary from year to year. Type 3 corresponds to fires which spread rapidly over wide areas. Such fires are typical of pastoral land management. The areas affected by fires of this type change relatively little from year to year. The way in which the different types of fires evolve provides information on their nature.

## 5. Conclusions

Processing the MODIS data—combined with a fairly strong empirical understanding of the terrain, local practices, and the types of landscape present—has enabled us to identify three broad patterns of wildfire distribution, defined by the number of active fires detected and the extent of burned areas. This typology can be correlated to the characteristics of the landscape and different fire practices. This study shows the potential in using fire data to differentiate geographic areas with different characteristics. MODIS fire products, active fires, and burned areas are proving to be good tools for spatial distinction. These initial results allow us to determine the functional characteristics of different patterns of fire and identify the spaces affected. In some areas, fire patterns have remained relatively stable over the 18 years spanned by our study. We also identified those areas that witnessed the biggest changes during this period.

Our approach was focused on identifying the occurrence and spread of fires, and tracking their interannual evolution. By dividing the zone into cells we were able to identify areas in which fire patterns were stable or variable. We are now working to build up a more accurate profile of the vegetation in these fire-affected areas. A further study focusing on specific cells and using data with a higher resolution would be useful in this respect, enabling us to analyze the relationship between fire patterns and vegetation. Plans are being made for an approach based on the analysis of spatial forms and structures using landscape metrics, allowing us to identify the key characteristics of these different fire patterns in greater detail. This would also help us to better understand the reasons behind changing patterns and functions. The methodology applied here to South-western

Madagascar would not necessarily be pertinent for other parts of the country. Madagascar is home to a large and diverse array of fire practices, particularly in the highlands and the east of the island, and our method may not be appropriate in those areas.

It seems highly likely that the three patterns of fire identified here also differ in terms of their environmental impact. This raises questions regarding the suitability of fire-fighting measures that fail to distinguish between different types of fires. Outlawing all types of fire appears to be entirely ineffective as a preventive measure. It is also ineffective as a means of protecting the environment, and fosters resentment among rural communities who use fire with considerable expertise and skill. It is also worth noting that, in spite of such repressive measures imposed by the government, the total surface area of burned zones has not decreased over the past 20 years. Adopting a differentiated approach to fires would allow for more rational management of such practices. By improving our understanding of fire patterns and their causes in south-west Madagascar, our hope is that this study will contribute to the creation of an effective strategy of fire management as recommended by the Food and Agriculture Organization of the United Nations (FAO) [66]. The results of this study must be further explored in different directions, in particular by identifying and comparing the different types of vegetation concerned, generally savannas and woodlands.

**Author Contributions:** L.C. analyzed the data and wrote and revised the manuscript. A.N.T. and A.B. checked and revised the manuscript. All authors have read and agreed to the published version of the manuscript.

**Funding:** This work was jointly supported by the University of Angers, LETG UMR 6554 CNRS, ESO UMR 6590 CNRS and Institut Universitaire de France.

**Institutional Review Board Statement:** Not applicable.

**Informed Consent Statement:** Not applicable.

**Data Availability Statement:** Not applicable.

**Acknowledgments:** This article is part of an ongoing doctoral research. The authors thank the members of the thesis committee, Félicitée Rejo-Fienena and Sébastien Caillault, for their fruitful discussions.

**Conflicts of Interest:** The authors declare no conflict of interest.

## Abbreviations

The following abbreviations are used in this manuscript:

| | |
|---|---|
| DEF | Water and Forestry Directorate |
| DREFF | Regional Directorate for the Environment Ecology and Forests |
| EOS | Earth Observing System |
| MODIS | Moderate Resolution Imaging Spectroradiometer |
| SEVIRI | Spinning Enhanced Visible Infrared Imager |
| MSG | Meteosat Second Generation |
| VIIRS | Visible Infrared Imaging Radiometer Suite |
| SLSTR | Sea and Land Surface Temperature Radiometer |
| GOES | Geostationary Operational Environmental Satellites |
| AVHRR | Advanced Very High Resolution Radiometer |
| VI | Vegetation Index |
| QGIS | Quantum Geographic Information System |
| GLM | Generalized Linear Model |

## Appendix A. Processing Chain

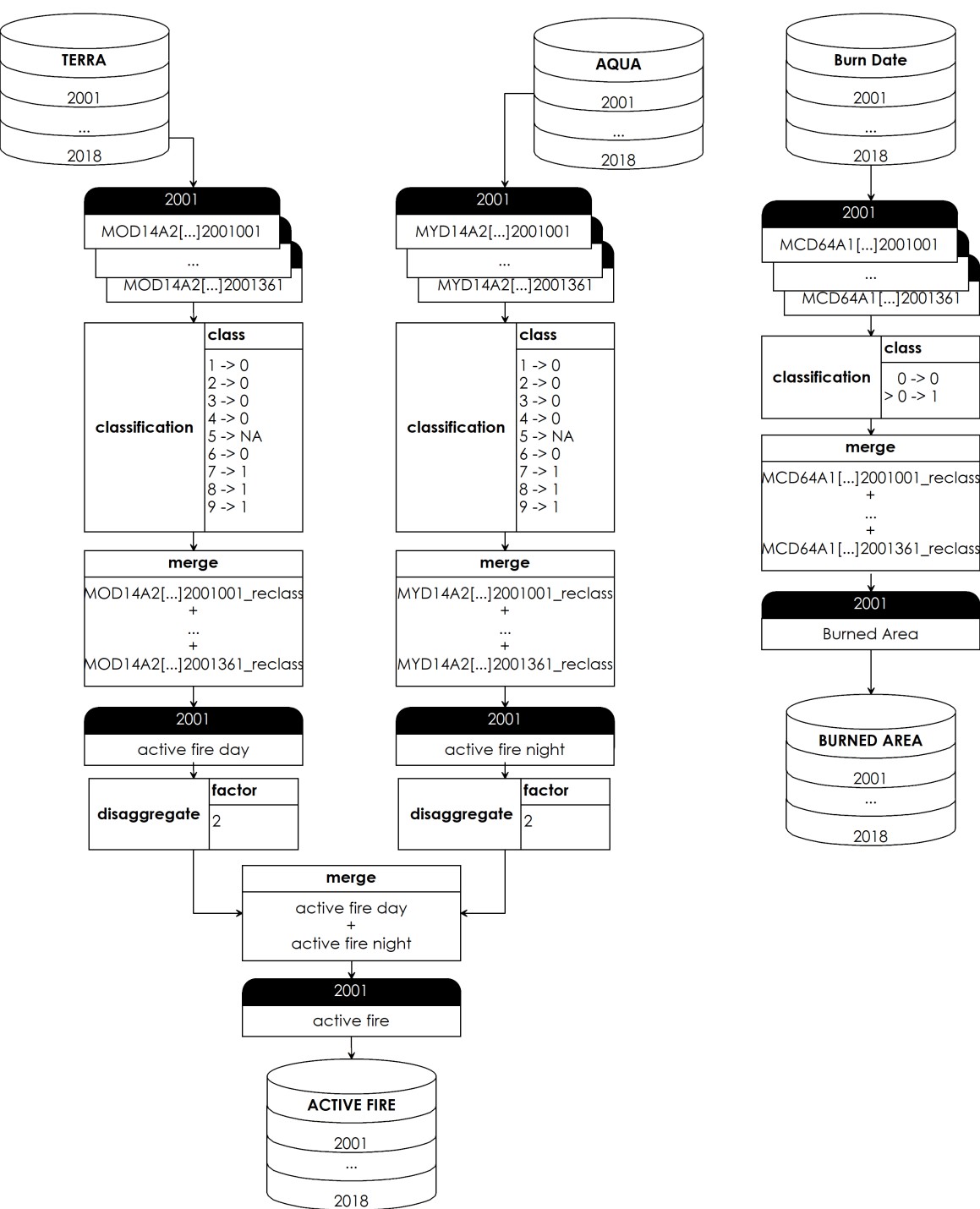

**Figure A1.** Pre-processing of MODIS active fire and burned area data.

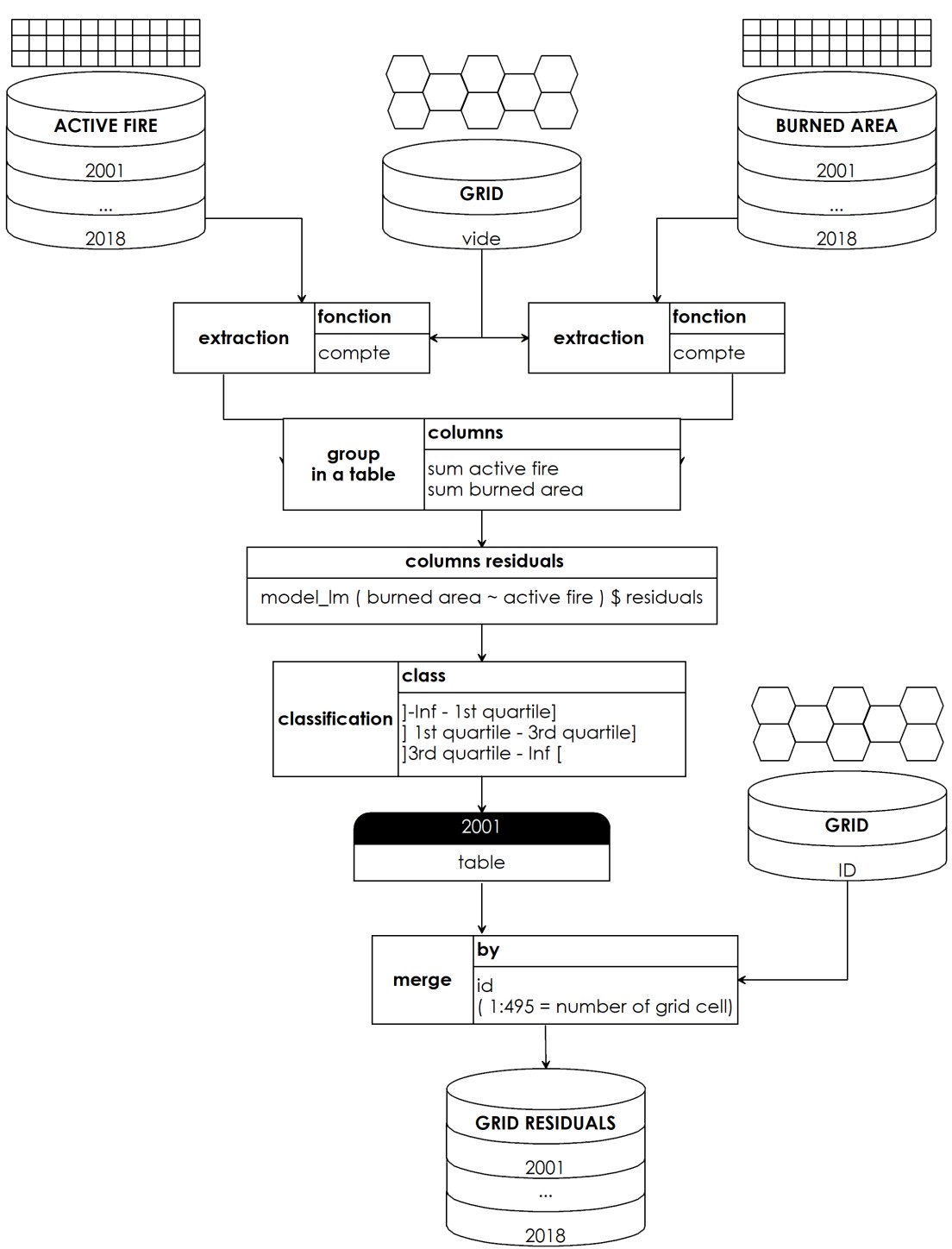

**Figure A2.** Processing of MODIS data.

## Appendix B. Result Tables for Part 3

**Table A1.** Results of calculated Chi2 or $\chi^2$, and Table showing $\chi^2$ values from 2001 to 2018.

| Year | Calculated $\chi^2$ | $\chi^2$ Table | Difference |
|------|---------------------|----------------|------------|
| 2001 | 14,326.04 | 11,162.93 | 3163.11 |
| 2002 | 24,348.71 | 22,523.26 | 1825.45 |
| 2003 | 34,851.47 | 32,462.27 | 2389.20 |
| 2004 | 23,291.59 | 19,129.89 | 4161.70 |
| 2005 | 32,729.10 | 30,222.56 | 2506.54 |
| 2006 | 25,433.87 | 21,495.20 | 3938.67 |
| 2007 | 35,194.89 | 32,529.71 | 2665.18 |
| 2008 | 29,599.58 | 26,351.77 | 3247.81 |
| 2009 | 25,964.71 | 22,059.65 | 3905.06 |
| 2010 | 15,484.31 | 12,374.92 | 3109.39 |
| 2011 | 25,058.82 | 20,572.80 | 4486.02 |
| 2012 | 22,021.18 | 19,329.57 | 2691.61 |
| 2013 | 26,433.91 | 22,749.01 | 3684.90 |
| 2014 | 23,945.61 | 19,676.46 | 4269.15 |
| 2015 | 25,310.35 | 21,333.92 | 3976.43 |
| 2016 | 29,570.94 | 26,375.94 | 3195.00 |
| 2017 | 23,044.81 | 20,049.53 | 2995.28 |
| 2018 | 18,783.24 | 15,659.25 | 3123.99 |

**Table A2.** Tschuprow coefficients 2001–2018.

| Tschuprow Coefficient | | | | | | | | |
|------|------|------|------|------|------|------|------|------|
| **2001** | **2002** | **2003** | **2004** | **2005** | **2006** | **2007** | **2008** | **2009** |
| 0.6313034 | 0.6739318 | 0.7384760 | 0.7005906 | 0.7255860 | 0.7078400 | 0.7375973 | 0.7172418 | 0.7177674 |
| **2010** | **2011** | **2012** | **2013** | **2014** | **2015** | **2016** | **2017** | **2018** |
| 0.6802232 | 0.7114308 | 0.6774802 | 0.7124084 | 0.6998994 | 0.6985112 | 0.7228292 | 0.6827748 | 0.6870120 |

**Table A3.** Coefficients of determination for the years 2001 to 2018.

| Coefficient of Determination | | | | | | | | |
|------|------|------|------|------|------|------|------|------|
| **2001** | **2002** | **2003** | **2004** | **2005** | **2006** | **2007** | **2008** | **2009** |
| 0.38 | 0.27 | 0.36 | 0.23 | 0.35 | 0.27 | 0.38 | 0.35 | 0.29 |
| **2010** | **2011** | **2012** | **2013** | **2014** | **2015** | **2016** | **2017** | **2018** |
| 0.31 | 0.32 | 0.38 | 0.30 | 0.25 | 0.48 | 0.45 | 0.41 | 0.36 |

## Appendix C

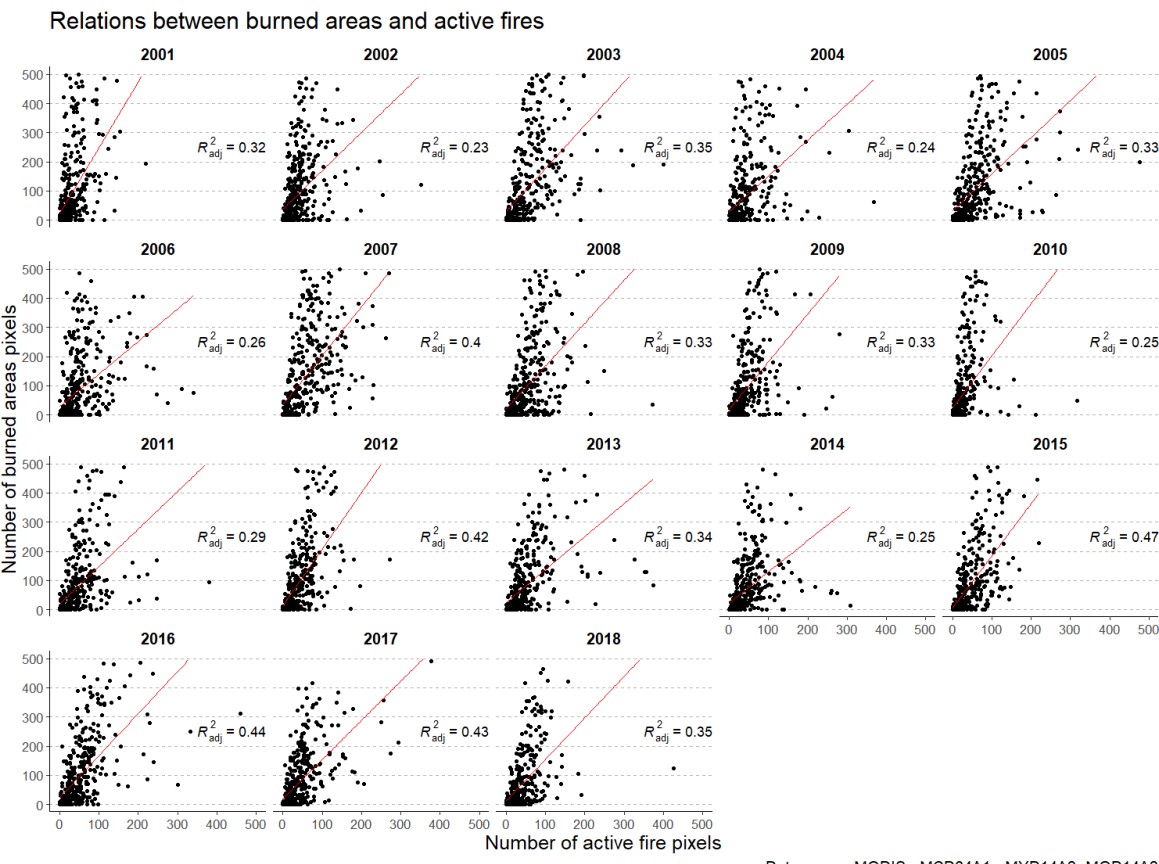

**Figure A3.** Correlation between burned areas and active fires.

## Appendix D

**Figure A4.** Mapping the residuals for each year between 2001 and 2018.

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
