# Peer review of "Spatial Analysis of the Occurrence and Spread of Wildfires in Southwest Madagascar"

_fire, doi:10.3390/fire5040098_

Round 1

Reviewer 1 Report

I thank the authors for their explanations regarding my previous reviews. I think the manuscript now clears up almost all my doubts. I hope that the process followed will also be clear to the average reader of this journal. With these considerations, I understand that the manuscript can now be published in Fire.

Author Response

I would like to thank the reviewer for his recommendations which have improved the paper considerably. 

Reviewer 2 Report

Authors have improved the paper by taking into account my previous comments. However,  the description  of satellite systems used to detect fires is  still approximate, and the relative literature is scarsely referenced. Moreover, the paper structure, in my opinion, still requires revisions.  Due to those weaknesses, I do not recommend the paper for publication.

Author Response

We have changed the description of satellite systems used to detect fires. Line 76-89. The paper structure has been revised to comply with the reviewers' requests. We hope that these changes will relieve your doubts. 

Reviewer 3 Report

Reviewer’s Report on the manuscript entitled:

A spatial analysis of the occurrence and spread of wildfires in SW Madagascar. Identifying and locating different types of burned areas.

The authors investigated the occurrences and spread of wildfires in southwest Madagascar using MODIS data. Although the topic and results are interesting and useful, the presentation of the manuscript must be improved.  

The title has an issue. Please change the title to

Spatial analysis of the occurrence and spread of wildfires in southwest Madagascar

Line 33. Please also say:

In addition to climate factors, topographic variables, such as altitude, slope, and aspect, can also affect the behaviour and speed of forest fires’ spread [https://doi.org/10.3390/su14073881]

Line 94. Needs a couple references for cloud effects. Please insert the following references:

https://doi.org/10.3390/rs12152446

https://doi.org/10.3390/rs12182870

Line 118. Please define NOAA. All abbreviations must be defined the first time they appear in the text and their style must be consistent. Please also include the missing acronyms in the acronym table at the end of the manuscript in line 638.

Lines 129, 231. Style issue. Please insert a comma after “i.e.”

Line 132. By referring to the Section number, please mention how the rest of the manuscript is organized.

Figures 1 and 2. The figures may have a copyright issue. Please create your own maps where applicable. Please also note that the font size of the texts and numbers in the figures should approximately have the same size as the font size of the figure caption, consistent and readable.

Section 2.1 should be called “Study region”

Section 2.4 should be called “Methods”

Figure 3 is not visible on the page. Please use geographic latitudes and longitudes when you display all the maps. See the first article I suggested above as an example.

Figure 4. Please insert the R^2 value in each panel.

Table 3. Format issue. Square brackets “]” should be in this format [xxxx]. Instead of ]xxxx[ use (xxxx) or instead of ]xxxx] use (xxxx], etc.

Figure 5 has lots of issues. The font size is not consistent. Some texts are relatively so large. No need to insert the latitudes and longitudes, etc.

Overall, the figure qualities are very poor and need extensive revisions.

Equation (2). Why does n come with two dots? Please remove the double dots that appeared as subscripts for n.

The are many redundant texts that can be removed. Please minimize the repetition and make the manuscript as concise as possible, so the main message is clearly mentioned to the readers.

Thank you

Regards,

Author Response

Thank you for the suggested changes. We have taken them all into account in order to improve the paper. 

Figure 1 & 2 : All maps and figures were produced by the authors. The data used are sourced in accordance with the regulations.

Figure 5 : All maps from 2001 to 2018 are available in Appendix D.

The text indeed contains some redundancies. Due to some misunderstandings by reviewers, we have chosen to emphasise some crucial points.

Round 2

Reviewer 2 Report

I found the paper improved in comparison with previous versions. However, both manuscript and figures still require revisions.

In the following some comments/suggestions for the authors. 

Line 78: Sensors having infrared channels..

Line 78-81: remove the sentence

Line 81: Since 1999, with the launch of Terra satellite of the EOS (..) program, MODIS (..) is used to monitor and characterize the fire processes (add references here)

Line 88-89: the sentence should be rearranged as: “Other authors exploited the 15 min temporal resolution of SEVIRI (), aboard MSG () geostationary satellites, to monitor fires with an increased frequency of observation. More recent instruments, such as VIIRS () and SLSTR () offering data at higher spatial resolution in the infrared bands (e.g., up to 375 m for VIIRS), have futher improved the detection capabilities of active fires (please, add references for SEVIRI, VIIRS and SLSTR).

Line 111: provide a reference for the MODIS algorithm

Line 112: hot volcanic features and gas flares may also lead to false positives

Line 113: remove the sentence

Line 116: “Due to the aformentioned issues, …

Line 119-123: remove the sentence

Line 123: I suggest to change the sentence as: “In this work, we combine information on active fires and burned areas to identify different fire spread patterns in the South-western Madagascar”;

Line 128-129: this sentence may be removed.

Line 169: In this study, we used both Aqua and Terra-MODIS data. Terra satellite…

Line 175: mentioned bands should be specified in table 2; those used by the MODIS algorithm could be marked in red.

Line 183: this information (spatial resolution) is already reported in text.

Line 240-244: this sentence should be reformulated.  You could modify the sentence as: “Fires of short-term duration may be undetected by MODIS owing to the daily temporal coverage”.

Line 260-261:.. confidence level; remove the rest of the sentence

Line 305-309: remove the sentence.

Line 341: you can remove “in order”

Line 366: see previous comment

Line 398: as previous comment

Line 503: top panel is shown in the appendix; it may be removed from Figure 5;

Line 523: region with major changes in the fire pattern should be highlighted in Figure 7

Line 537: retrieved from..

Line 550: these fires correspond to…; They reflect

Appendix C: labels are difficult to read;

Author Response

Thank you for your attention to our paper. Your comments were very thought-provoking and helped us to improve our text. The readability of figures and legends have been improved. We have made all the suggested changes except line 128-129 : As this sentence explains one element specific of this work, we prefer not to delete it.

Reviewer 3 Report

I would like to thank the authors for addressing my comments. In my view, the manuscript can be accepted.

Thank you for your contribution!

Author Response

We thank the reviewer for his pertinent comments and his approval for publication.

This manuscript is a resubmission of an earlier submission. The following is a list of the peer review reports and author responses from that submission.

Round 1

Reviewer 1 Report

The re-submitted manuscript has addressed all comments adequately.

Reviewer 2 Report

I regret to find no significant changes in this new version of the manuscript. Nor convincing clarifications in the authors' reply notes concerning my two previous revisions.

Some bibliographic references remain with statements that I suggested changing in my previous review. For example:

Lines 185-186: "Both products "active fires" and "burned area" can be complementary interrelationship used [15]. "(See second paragraph of my previous revision).

In addition, the authors continue to make inconceivable basic errors that prevent this article from being published in a journal such as Fire. For example, on lines 119-120, they write: "Visible Infrared Imaging Radiometer Suite (VIIRS) fire data from the Landsat satellites". Can the authors explain how such a naive mistake can be made, putting together a sensor and a satellite that have nothing to do with each other?

I recognize that the authors seem to have mastered the statistical techniques applied to the products used to achieve their objective. However, they demonstrate a great unfamiliarity with the products themselves, the sensors that have registered these images and the platforms (satellites) on which they are mounted.

Finally, I would like to ask the authors once again for clarification regarding the two products used in this study (MCD64A1 burned area product and the MOD14A2 (day) and MYD14A2 (night) active fire products). The MCD64A1 product uses, as they clarify in the manuscript, the MOD14A2 (day) and MYD14A2 products. From an exclusively statistical point of view, what is the point of analyzing or studying correlations between both products if the first MCD64A1 (area burned) implicitly includes the second (MOD14A2 and MYD14A2) of active fires?

Reviewer 3 Report

The Authors have substantially replied positively to my comments.

There are some further remarks

Line 119-120 “ … the Visible Infrared Imaging Radiometer Suite (VIIRS) fire data from the Landsat satellites”  VIIRS instrument is aboard the joint NASA/NOAA Suomi National Polar-orbiting Partnership (Suomi NPP) and NOAA-20 satellites, and not Landsat satellites.

Line 221 “ … are employs to derived” à  … are employed to derive

Line 306 “But the sensors cannot hope to detect all fires”. Please eliminates this sentence.

Line 308  “… prove to be too small for detection.” You may add this recent paper Ramo et al., 2021 “African burned area and fire carbon emissions are strongly impacted by small fires undetected by coarse resolution satellite data” PNAS 2021 Vol. 118 No. 9 e2011160118

Line 312 “However, we do believe that observing …” à However, observing …